# PGC-1α activation to enhance macrophage immune function in mycobacterial infections

**Joel R. Frandsen**[1], **Zhihong Yuan**[1], **Brahmchetna Bedi**[2], **Zohra Prasla**[3], **Seoung-Ryoung Choi**[4], **Prabagaran Narayanasamy**[4], **Ruxana T. Sadikot**[1]*

1 Division of Pulmonary, Critical Care & Sleep, Department of Internal Medicine, University of Nebraska Medical Center, Omaha, Nebraska, United States of America, 2 Division of Infectious Diseases, Department of Medicine, Emory University, Atlanta, Georgia, United States of America, 3 Pulmonology and Critical Care Department, University of California at Los Angeles, Los Angeles, California, United States of America, 4 Department of Pathology and Microbiology, College of Medicine, University of Nebraska Medical Center, Omaha, Nebraska, United States of America

* rsadikot@unmc.edu

**Data Availability Statement:** All relevant data are within the paper and its Supporting information files.

## Abstract

Nontuberculous Mycobacteria (NTM) are a heterogeneous group of environmental microorganisms with distinct human pathogenesis. Their incidence and prevalence are rising worldwide, due in part to elevated antimicrobial resistance which complicates treatment and potential successful outcomes. Although information exists on the clinical significance of NTMs, little is known about host immune response to infection. NTM infections alter macrophage mitochondrial capacity and decrease ATP production, efficient immune response, and bacterial clearance. Transcription factor peroxisome proliferator activated receptor (PPAR) γ coactivator-1α (PGC-1α) is a master regulator of mitochondrial biogenesis, influencing metabolism, mitochondrial pathways, and antioxidant response. Mitochondrial transcription factor A (TFAM) is a protein essential for mitochondrial DNA (mtDNA) genome stability, integrity, and metabolism. Both PGC-1α and TFAM regulate mitochondrial biogenesis and activity, and their disruption is linked to inflammatory signaling and altered macrophage function. We show that NTM causes macrophage mitochondrial damage and disrupted bioenergetics. Mechanistically we show that this is related to attenuation of expression of PGC-1α and TFAM in infected macrophages. Importantly, rescuing expression of PGC-1α and TFAM using pharmacologic approaches restored macrophage immune function. Our results suggest that pharmacologic approaches to enhance mitochondrial function provide a novel approach to target macrophage immune function and means to combat NTM infections.

## Introduction

Bacterial species are constantly evolving and developing methods to evade host immunity, with resistance to antibiotics becoming an ever increasing burden on healthcare [1–3]. To combat this, researchers have shifted to non-traditional pathways and mechanisms to target infectious bacteria [4, 5]. This has led to an increased focus on host-directed therapies that

**Funding:** This research was funded through grants from the U.S. Department of Veterans Affairs (BX001786), and National Institutes of Health/The National Heart, Lung, and Blood Institute (R01-HL144478) The funders had no role in study design, data collection and analysis, decision to publish, or preparation of the manuscript.

**Competing interests:** The authors have declared that no competing interests exist.

target immune cell function rather than the bacteria itself [6, 7]. Nontuberculous mycobacteria (NTM) are a class of mycobacterium ubiquitously prevalent in the environment and are frequently encountered by humans [8, 9]. The incidence and prevalence of NTM diseases are rising worldwide, particularly in immunocompromised individuals living with HIV and patients with chronic lung diseases—including cystic fibrosis (CF), COPD, and non-CF bronchiectasis [10–12].

The majority of NTM infections arise from *Mycobacterium avium complex* (MAC) and *Mycobacterium abscessus* (MAB) [8, 9]. *MAB* is an especially difficult bacterial species to treat due to multiple mechanisms of immune system avoidance [10–13]. The high level antimicrobial resistance displayed by these organisms complicates treatment and potential successful outcomes, with treatment consisting of a lengthy pharmaceutical regimen utilizing a cocktail of antibiotics for up to 24 months [14, 15]. While much information exists on the clinical significance of NTM, relatively little is known about the host immune response to NTM infection.

Alveolar and lung tissue macrophages are the first line of defense against NTM infections, initiating the innate immune response during the initial infection, involving identification of bacterial pattern recognition receptors (PRR), phagocytosis, and subsequent lysosomal degradation [16]. Macrophages are the primary host cells that initiate an immune response to NTM [17]. Mycobacteria are well recognized for their remarkable ability to sense engulfment by macrophages and then survive the numerous and different forms of macrophage bactericidal attack [8, 9, 13, 18]. NTMs possess several unique features that allows it to reside in the macrophages and colonize airways, including the ability to prevent maturation and acidification of phagolysosome complexes, and the ability to form large, extracellular clumps and serpentine cords of bacteria that are too large to be phagocytosed by macrophages [10, 11, 13, 17, 19, 20]. Upon ingestion by macrophages, NTMs can arrest lysosomal degradation and actively replicate inside phagosomes [10, 11, 19, 21]. These bacteria can also alter and influence immune cell energetics, damaging mitochondria and reducing their metabolic capacity, leading to decreased antibacterial activity of macrophages [13, 19, 22].

Efficient bacterial killing requires a high rate of metabolic activity, making macrophage mitochondrial activity a key component in bacterial clearance and immune cell function [16, 23, 24]. Peroxisome proliferator activated receptor (PPAR) γ coactivator 1 α (PGC-1α) is a transcription factor that functions as a master regulator of mitochondrial biogenesis and influences genes involved in energy metabolism and mitochondrial pathways [25, 26]. Induction of PGC-1α orchestrates the transcription and replication of mitochondrial DNA (mtDNA) components involved in metabolism, and functions to fine tune the balance of mitochondrial fusion and fission during the process of mitochondrial biogenesis. In addition to metabolic activity, PGC-1α also actively participates in protection against inflammatory cellular environments and excessive production of reactive oxygen species (ROS) through regulation of cellular antioxidant response [27, 28]. PGC-1α influences immune response by regulating mitochondrial energetics of macrophages and other immune cells involved in responding to bacteria and pathogen infection [26, 29, 30]. The presence and activity of PGC-1α is indispensable in cellular functions responding to infection by NTMs and other pathogens.

In this study, we investigate the impact of NTM infections on host macrophage mitochondria. We hypothesize that increasing mitochondrial biogenesis through increased PGC-1α expression will enhance macrophage-mediated clearance of NTMs through modulation of mitochondrial energetics. We utilized RAW 264.7 macrophages and human THP-1 monocytes for infection by MAB and MAC and evaluated different measures of macrophage activity and mitochondrial function to elucidate the relationship between PGC-1α and host immune response. Our data show that NTM disrupt the ability of macrophages to generate functional

mitochondria by attenuating PGC-1α and TFAM. Most importantly, PGC-1α pharmacologic activators ZLN005 and metformin enhance host immune cell response to NTM infection [31, 32]. Through our research, we provide evidence that PGC-1α activation and expression is integral to immune cell function and metabolism, and is intimately correlated to bactericidal activity of macrophages in clearing NTM infections.

## Materials and methods

### Reagents and chemicals

The following primers were acquired from Thermo Fisher (Waltham, MA) and used for PCR: *ppargc1* (Mm01208835_m1, hs01016719), *tfam* (Mm00447485_m1, Hs00273372_s1), *gapdh* (Mm99999915_g1, hs02786624). The following primary and secondary antibodies were used: PGC-1α (Cell Signaling Technology (CST), Danvers, MA 2178), PGC-1α (Santa Cruz Biotechnology (SCBT), Dallas, TX sc-518025), TFAM (CST, 8076), mtTFA (SCBT, sc-166965), β-Actin (CST, 3700), β-Tubulin (CST, 2146) Anti-rabbit IgG HRP (CST, 7074), Anti-mouse IgG HRP (CST, 7076), Goat anti-Rabbit IgG AF 488 (Invitrogen, Waltham, MA, A11008), Donkey anti-Mouse IgG AF 647 (Invitrogen, A31571). MitoTracker Green FM (MTG) (Invitrogen, M7514), Tetramethylrhodamine Methyl Ester (TMRM) (Invitrogen, T668), MitoSOX Red (Invitrogen, M36007) 4', 6-diamidino-2-phenylindole, dihydrochloride (DAPI) (CST, 4083), Hoechst 33342 (CST, 4082). Unless otherwise specified, reagents and chemicals were obtained through Sigma Aldrich or Thermo Scientific.

### Cell culture

RAW 264.7 mouse macrophages (TIB-71 ATCC, Manassas, VA) and THP-1 (TIB-202, ATCC) human monocytes were utilized for infection experiments. RAW 264.7 cells were maintained in DMEM containing 10% fetal bovine serum (FBS) (Gibco, Grand Island, NY) and 1% Penicillin-Streptomycin (Gibco), and cultured in T75 flasks with half media changes every two days. Upon reaching confluence culture media was aspirated, and the flask gently washed with PBS. Cells were dissociated using a 0.25% trypsin solution (Gibco) and a cell scraper. Collected cells were added to a 15 mL conical tube containing an equal volume of regular culture media and centrifuged at 1300 x g for 5 minutes. The media was aspirated, pellet resuspended in 1 mL of culture media, and cell culture solution was filtered through a 100 μm nylon cell strainer (Falcon, Corning, NY). Cell counting was determined using a Cell Countess II (Invitrogen). Cells were then seeded in tissue culture plates at appropriate densities (6-well plate: $1 \times 10^6$ cells/well; 12-well plate: $4 \times 10^5$ cells/well; 24-well plate: $1.5 \times 10^5$ cells/well; 96-well plate: 7,000 cells/well).

THP-1 monocytes were maintained in RPMI containing 10% FBS, 1% Penicillin-Streptomycin, 1%, 1% GlutaMAX (Gibco), and 1% HEPES [5 M] (Gibco). Cells were cultured in T75 flasks with half media changes every three days until reaching confluence. For macrophage differentiation, normal culture media was removed, flask gently washed with PBS, and culture media containing 30 ng/mL phorbol 12-myristate 13-acetate (PMA) (Sigma Aldrich) was added to cells for 48 hours. PMA media was removed and replaced with normal culture media, and cells were allowed to rest 24 hours prior to treatment or infection.

### Drug treatment

ZLN005 (Cayman Chemicals, Ann Arbor, MI) was dissolved in DMSO and used to treat cells at a final concentration of 2 μM. Metformin (Selleck Chemicals, Houston, TX) was dissolved in water and used at a final concentration of 2 mM in cell cultures. SR-18292 (SR) (Selleck

Chemicals) was dissolved in DMSO and used to treat cells at a final concentration of 10 μM. Cultured cells were incubated with treatment chemicals for 24 hours prior to infection. Toxicity of ZLN005 and metformin to both cell lines (S2) and bacteria (S3) was determined to be significantly higher than the *in vitro* concentrations used in this manuscript.

## Bacterial cultures

*Mycobacterium abscessus* (19977 ATCC) and *Mycobacterium avium* strain MAC 101 (700898 ATCC) were grown on 7H9 agar or liquid media containing a 10% solution of oleic acid, albumin, dextrose, and catalase (OADC) supplement (Sigma Aldrich). A single colony was isolated from the agar plate and added to a 50 mL vented-cap conical tube containing 20 mL 7H9 media and grown in a rotating incubator at 200 RPM 37°C for 4–5 days. Inoculated cultures were grown to $OD_{600}$ of ~0.600 as measured by a BioMate 3S UV-Visible Spectrophotometer (Thermo Scientific). Inoculated cultures were centrifuged at 12,000 x g for 12 minutes. The bacterial pellet was then resuspended in PBS, recentrifuged, and stored at -80°C.

## Infection

RAW 264.7 and THP-1 cells were cultured as previously described. Prior to infection, normal culture media was removed, wells were gently washed with PBS, and antibiotic-free media was added to cells. Through $OD_{600}$ measurements and CFU counting, bacterial numbers were determined to be $2.4 \times 10^8$ CFU/ml at $OD_{600}$ (1.0) [33]. The corresponding numbers were added to wells at MOIs of 1 and 10, and infected for 6 or 24 hours prior to collection for analysis.

## SDS-PAGE Western blot

Media was removed from 6-well plates, and wells were washed three times with ice-cold PBS. Cell Lysis Buffer (10X) (CST) with the addition Halt Protease & Phosphatase Inhibitor Single-Use Cocktail (100X) (Thermo Scientific) was added to wells at a volume of 100 μL and allowed to gently rock on ice for 15 minutes. Cell lysates were scraped from wells, collected in microcentrifuge tubes, centrifuged at 15,000 RPM for 10 minutes, and supernatant was collected. Protein concentration was determined using a Pierce BCA assay (Thermo Fisher) and compared to a standard curve of BSA concentrations. Cell lysates were combined with Laemmli 4X Buffer (Sigma Aldrich) and heated at 95°C for 5 minutes. The boiled lysate mixtures were loaded into a 4–15% Mini-PROTEAN TGX SDS PAGE gel (BIO-RAD, Hercules, CA) at a concentration of 30 μg of protein per lane, with Precision Plus Protein Dual Color Standards (BIO-RAD). Gels were resolved in Tris/Glycine/SDS buffer (BIO-RAD) and run at 125 V for 45 minutes on a PowerPac energy supply (BIO-RAD). Gels were removed from cassettes, washed in PBS, and transferred to PVDF membrane in Tris/Glycine buffer (BIO-RAD) at 65 V for 80 minutes. Presence of protein on blots was determined using Ponceau S stain (Sigma Aldrich). Blots were washed in PBS, and blocked with PBS containing 5% BSA (Sigma Aldrich) for 1 hour. Blots were washed in PBS, and incubated with primary antibody in PBST containing 5% BSA overnight at 4°C. Blots were washed in PBS, incubated with the appropriate HRP-conjugated secondary antibody for 1 hour at RT, and washed again. Blots were imaged using SuperSignal Western Dura Substrate (Thermo Fisher), and chemiluminescence detected using a ChemiDoc MP system (BIO-RAD). Densiometric analysis was performed with ImageJ (Madison, WI). Raw Western blot images are included in the supporting information (S1 Raw images).

## Total RNA extraction

Total RNA was extracted using the RNeasy Mini Kit (Qiagen, Hilden, Germany) according to manufacturer's instructions. Cells were cultured, treated, and infected as previously described. Media was removed from 12-well plates and wells were washed three times with ice-cold PBS. Cells were lysed using 350 μL of RLT Buffer and 350 μL of 70% ethanol per well and mixed by pipetting. The 700 μL volume was transferred to a RNeasy Mini spin column in a 2 mL collection tube, and centrifuged 15 seconds at 8000 x g, with the liquid flow-through discarded. 700 μL of RW1 Buffer was added to each spin column and recentrifuged, with the flow-through discarded. 500 μL RPE Buffer was added to the spin column and recentrifuged, with the flow-through discarded. Another 500 μL RPE Buffer was added to the spin column, and was centrifuged for 2 minutes at 8000 x g. The RNA was eluted by adding 30 μL of RNase-free water to the spin column membrane and centrifuged for 1 minute at 8000 x g. The concentration of RNA was determined using a NanoDrop One C (Thermo Fisher)

## RT-PCR

RT-PCR was performed using SuperScript IV Reverse Transcriptase (Invitrogen) according to the manufacturer's instructions. Briefly, 1 μg of total RNA was added to a mixture of Oligo $(dT)_{12-18}$ and dNTP Mix, and incubated in a T100 Thermal Cycler (BIO-RAD). Following a brief incubation on ice, 5X First Strand Buffer, DTT, and RNaseOUT were added to each microcentrifuge tube and heated in the thermal cycler at 42˚C for 2 minutes. SuperScript IV RT was added to the mixture and the samples incubated at 42˚C for 50 minutes, and then incubated at 70˚C for 15 minutes to inactivate the reaction. The synthesized cDNA was then used for PCR. cDNA was added to TaqMan Universal PCR Master Mix (Applied Biosystems, Waltham, MA) with the corresponding TaqMan primers (Applied Biosystems) including *ppargc1a* (mm00447183, hs01016719), *tfam* (mm00447485, hs00273372), and *gapdh* (mm99999915, hs02786624) and run through the PCR program on a 7500 Real Time PCR System (Applied Biosystems).

## Immunocytochemistry

Cells were cultured, treated with metformin [2 mM], ZLN005 [2 μM], and/or SR [10 μM] and infected as previously detailed. Media was removed from 96-well plates and wells were washed with PBS three times for 5 minutes each. Cells were fixed with PBS containing 4% paraformaldehyde for 15 minutes, and washed with PBS three times for 5 minutes each. Cells were then permeabilized with PBS containing 0.3% Triton X-100 for 10 minutes, and washed with PBS three times for 5 minutes each. Cells were blocked with PBST containing 5% goat serum for 30 minutes at room temperature, and primary antibodies diluted in PBST containing 5% goat serum were added and plate was incubated overnight at 4˚C. Cells were washed with PBST three times for 5 minutes each, and the appropriate fluorescent-conjugated secondary antibodies in PBST containing 5% goat serum were added for 1 hour at room temperature. Cells were washed with PBST three times for 5 minutes each, counterstained with 1 μg/mL DAPI for 2 minutes, washed again with PBS three times for 5 minutes each. Cells were then imaged on a BZ-X800 confocal microscope (Keyence, Osaka, Japan). Relative fluorescence intensity was determined by averaging fluorescence to the number of DAPI positive cells.

## Intracellular killing assay

RAW 264.7 macrophages were seeded at a density of 150,000 cells/well in a 24-well plate, treated with metformin or ZLN005 for 24 hours, and infected with MAB (MOI 10) for either 6

or 24 hours. Media was aspirated, wells were gently washed twice with HBSS, and 100 μL of PBS containing Triton X-100 (1%) and SDS (0.1%) was added to wells. Cell lysates were collected with gentle scraping, and serially diluted in PBS to concentrations of $10^{-6}$, $10^{-7}$, $10^{-8}$. 30 μL of diluted lysate solution was added to 7H9 agar plates and spread across the plate using an L-shaped cell spreader. Plates were placed in an incubator at 37°C and 5% $CO_2$ atmosphere for 5–7 days, and bacterial colonies were counted and converted to CFU/mL(log).

## Phagocytosis assay

Effect of drug treatment on phagocytosis was performed using the Vybrant Phagocytosis Assay (Thermo Fisher) according to manufacturer's instructions. THP-1 cells were cultured in 6-well plates at a seeding density of $1.5 \times 10^6$ cells/well and grown to confluence. Cells were treated with metformin or ZLN005 for 24 hours prior to performing the assay. Culture media was aspirated, wells were washed with PBS, and 1 mL culture media was added to wells. Wells were gently scraped and collected for centrifugation (1200 g x 6 mins). Cell counts were determined as previously described, and DMEM was added to each tube to a final concentration of $10^6$ cells/mL. One vial of fluorescent particles and HBSS concentrate was thawed and 0.5 mL HBSS was added to particle vial and briefly sonicated to disperse particles. The suspension was added to a centrifuge tube containing 4.5 mL sterile $ddH_2O$ and sonicated to homogenize the mixture. Cells were added to a 96-well plate at a density of $1x10^5$ cells per well, allowed to adhere for one hour, and 100 μL of the fluorescent particle solution was added to each well and incubated for 2 hours. Media was removed from wells, and 100 μL of the trypan blue solution was added to quench extracellular fluorescence and allowed to incubate for one minute. Trypan solution was aspirated, and fluorescence (480 nm excitation/520 nm emission) was measured on a microplate reader.

## Mitochondrial ROS production

THP-1 cells were cultured, treated with metformin [2 mM], ZLN005 [2 μM], and SR [10 μM], and infected in 96-well plates (Grenier Bio-One) as previously described. Media was aspirated from a 96-well plate, and wells were gently washed with warm HBSS. MTG and MitoSOX Red (Invitrogen) were added to RPMI media without phenol red, and cells were incubated in the solution for 30 minutes. Media was removed, wells were washed with HBSS, and HBSS containing 1 μg/mL Hoechst was added to cells for 3 minutes. The solution was aspirated, wells were washed with HBSS, and RPMI media without phenol red was added to wells. Live cells were imaged on a BZ-X800 confocal microscope.

## Mitochondrial membrane integrity

THP-1 cells were cultured in 96-well plates (Grenier Bio-One), treated with metformin [2 mM], ZLN005 [2 μM], and SR [10 μM], and infected as previously described. Media was aspirated from a 96-well plate, and wells were gently washed with warm HBSS. MTG and TMRM were added to RPMI media without phenol red at a final concentration of 200 nM each, and cells were incubated in the solution for 30 minutes. Media was removed, wells were washed with HBSS, and HBSS containing 1 μg/mL Hoechst was added to cells for 3 minutes. The solution was aspirated, wells were washed with HBSS, and RPMI media without phenol red was added to wells. Live cells were imaged on a BZ-X800 confocal microscope.

## Statistical analysis

All experiments were performed in triplicate in at least 3 independent experiments and presented data is the mean +/- standard error. Data were analyzed for statistical significance using Student t-test or ANOVA with Excel (Microsoft, Redmond, WA) or PRISM (GraphPad, San Diego, CA).

# Results

## NTM infection induces mitochondrial damage and reduces number of mitochondria

Mitochondria are membrane-bound organelles involved in myriad biological processes that are the powerhouse necessary for cellular function [16, 23]. Given their significant role in cellular function, mitochondria are a common target for bacterial-derived effectors [34–38]. However, the impact of NTM on host mitochondria has not been well characterized. To investigate the effects of NTM on macrophage mitochondria we infected macrophages with MAC and MAB and evaluated markers of mitochondrial damage and mitochondrial numbers. Infection of macrophages with NTM resulted in increased mtDNA oxidative damage, as determined by the ratio of 79 bp mtDNA fragments to full length 230 bp mtDNA [39]. Both MAC and MAB infection resulted in a significant increase in the 79:230 ratio in macrophages compared to uninfected controls (Fig 1A). We also observed a significant decrease in mitochondrial number in MAC and MAB infected macrophages compared to control (Fig 1B). Mitochondria were quantified by randomly selecting areas and counting the number of mitochondria per high power field (hpf) for each condition and averaged. We also investigated the effect of NTM infection on mitochondria ROS production in macrophages. Infection with MAC induced production of superoxide in macrophages (Fig 1C), which rose significantly during longer infection periods (Fig 1D). These results suggests that NTM infection of macrophages results in mitochondrial damage with reduction in mitochondrial numbers which can impact the immune function of macrophages.

## ZLN005 increases PGC-1α and TFAM expression in NTM infected macrophages

Previously, we demonstrated that small molecules targeting PGC-1α (ZLN005) post-translational regulation rescue mitochondrial and metabolic derangements caused by *P. aeruginosa* infection [39]. Since NTM infections cause mitochondrial damage and attenuate the expression of PGC-1α we hypothesized that ZLN005 will be able to rescue the expression of PGC-1α and its downstream effector TFAM. RAW 264.7 macrophages were treated with ZLN005 prior to MAC infection to evaluate the effect on gene and protein expression of PGC-1α and TFAM. PCR analysis revealed ZLN005 treatment resulted in a significant increase in PGC-1α mRNA (*ppargc1a*) compared to control, and attenuated MAC-mediated decreases during 6-hour infection (Fig 2A) and 24-hour infection (Fig 2B). A similar trend was observed in TFAM mRNA (*tfam*), with ZLN005 treatment significantly increasing expression compared to controls and attenuating MAC-mediated decreases during 6-hour infection (Fig 2C) and 24-hour infection (Fig 2D).

Protein expression of PGC-1α and TFAM was evaluated through Western Blotting and immunocytochemistry. ZLN005 treatment significantly increased protein expression levels of PGC-1α (Fig 2E) and TFAM (Fig 2G) compared to controls, and blunted MAC-mediated decreases during a 6-hour infection period, as determined by densiometric evaluation of Western Blot bands (Fig 2I). We also observed a similar trend during longer infection times.

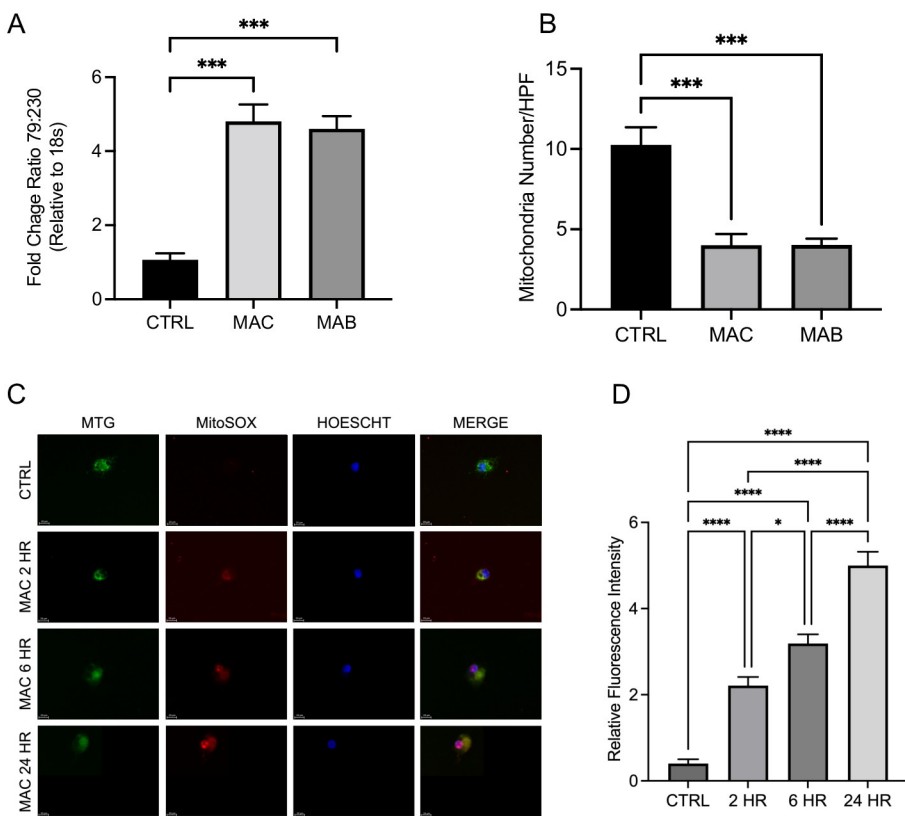

**Fig 1. NTM infection induces mitochondrial damage and reduces mitochondria number.** MAC and MAB infection in THP-1 macrophages induced significant elevation in mtDNA damage (A), and significantly reduced the number of mitochondria compared to uninfected controls (B). MAC infection induced mitochondrial damage through production of superoxide and ROS (C) that was significantly higher than uninfected control (D). Cells stained with MitoTracker Green (MTG) (green), MitoSOX (red), and Hoechst (blue); scale bar 20 μm. Mean +/- S.E.M shown, data representative of three independent experiments performed in triplicate. Statistical significance determined using ANOVA, with p values equaling *p<0.05, **p<0.01, ***p<0.005, ****p<0.001.

ZLN005 treatment significantly increased protein expression levels of PGC-1α (Fig 2F) and TFAM (Fig 2H) compared to controls, and attenuated MAC-mediated decreases during a 24-hour infection period as determined by densiometric evaluation of Western Blot bands (Fig 2J).

Representative images of the immunofluorescence staining for PGC-1α (Fig 2K) and TFAM (Fig 2L) are shown with quantification of relative intensity (Fig 2M and 2N, respectively). Together, these data show that activating PGC-1α by small molecule activator ZLN005 can rescue the expression of PGC-1α and its dependent gene TFAM in macrophages that are infected with NTM.

## Metformin increases PGC-1α and TFAM expression in NTM infected macrophages

Given ZLN005's beneficial effects on PGC-1α expression, we wanted to investigate whether a pharmacologic strategy using FDA-approved drugs that target enhancement of PGC-1α activity would rescue the attenuated expression found in NTM infected macrophages. Metformin is a widely used medication for treatment of Diabetes Mellitus, and exhibits the ability to

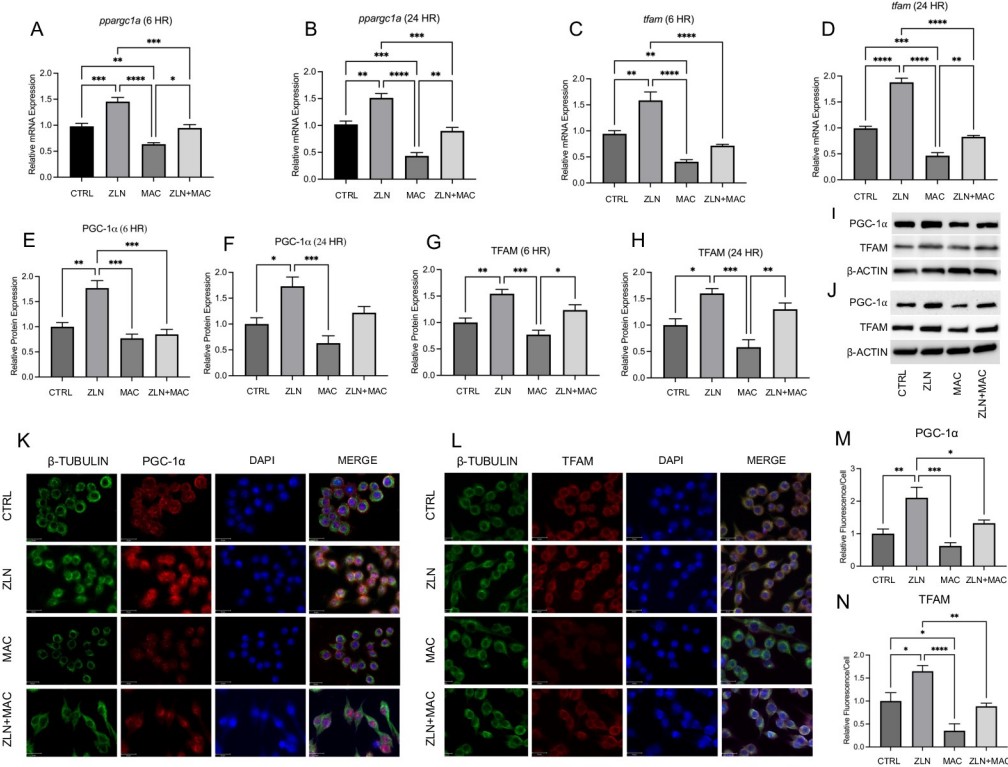

**Fig 2. ZLN005 increases PGC-1α and TFAM expression.** Relative mRNA expression of *ppargc1a* at 6 hours post infection (hpi) (A) and 24 hpi (B) and *tfam* at 6 hpi (C) and 24 hpi (D). Relative protein expression of PGC-1α at 6 hpi (E) and 24 hpi (F), and TFAM at 6 hpi (G) and 24 hpi (H), and representative blot images at 6 hpi (I) and 24 hpi (J). Representative immunofluorescence images of PGC-1α (K) and quantitation of relative fluorescence intensity of PGC-1α (M). β-Tubulin (green), PGC-1α (red), DAPI (blue); scale bar 20 μm. Representative immunofluorescence images of TFAM (L) and quantitation of relative fluorescence intensity of TFAM (N). β-Tubulin (green), TFAM (red), DAPI (blue); scale bar 20 μm. Mean +/- S.E.M shown, data representative of three independent experiments performed in triplicate. Statistical significance determined using ANOVA, with p values equaling *p<0.05, **p<0.01, ***p<0.005, ****p<0.001.

increase PGC-1α expression and activity through post-translational modifications including AMPK-mediated phosphorylation [40, 41]. RAW 264.7 macrophages were treated with metformin prior to MAC infection to evaluate the effect on gene and protein expression of PGC-1α and TFAM. PCR analysis revealed metformin treatment resulted in a significant increase in PGC-1α mRNA compared to control cells treated with vehicle alone. Most importantly, MAC-mediated decreases in PGC-1α during 6-hour infection (Fig 3A) and 24-hour infection (Fig 3B) were rescued by treatment with metformin. A similar trend was observed in the expression of TFAM. Metformin treatment significantly increased expression compared to controls and attenuated expression by MAC infection were restored during 6-hour (Fig 3C) and 24-hour infection (Fig 3D). Protein expression of PGC-1α and TFAM was evaluated through Western Blotting and immunocytochemistry. Metformin treatment significantly increased protein expression levels of PGC-1α (Fig 3E) and TFAM (Fig 3F) compared to controls, and blunted MAC-mediated decreases during a 6-hour infection period, as determined by densiometric evaluation of Western Blot bands (Fig 3I). Metformin treatment significantly increased protein expression levels of PGC-1α (Fig 3G) and TFAM (Fig 3H) compared to controls, and attenuated MAC-mediated decreases during a 24-hour infection period as

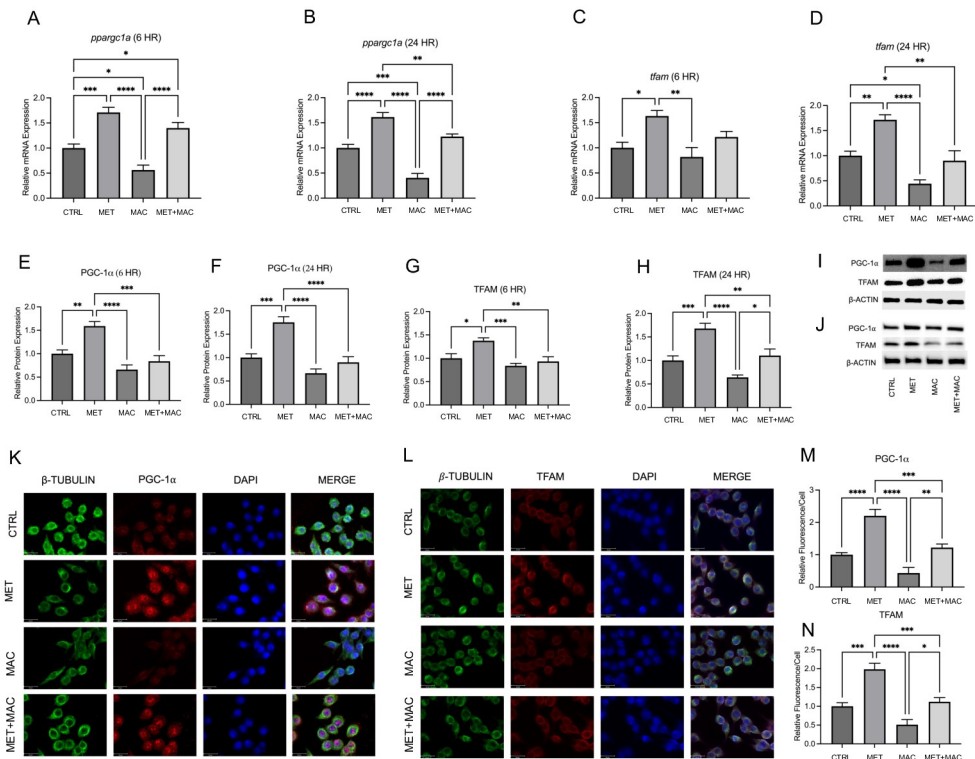

**Fig 3. Metformin increases PGC-1α and TFAM expression.** Relative mRNA expression of *ppargc1a* at 6 hpi (A) and 24 hpi (B) and *tfam* at 6 hpi (C) and 24 hpi (D). Relative protein expression of PGC-1α at 6 hpi (E) and 24 hpi (F), and TFAM at 6 hpi (G) and 24 hpi (H), and representative blot images at 6 hpi (I) and 24 hpi (J). Representative immunofluorescence images of PGC-1α (K) and quantitation of relative fluorescence intensity of PGC-1α (M). β-Tubulin (green), PGC-1α (red), DAPI (blue); scale bar 20 μm. Representative immunofluorescence images of TFAM (L) and quantitation of relative fluorescence intensity of TFAM (N). β-Tubulin (green), TFAM (red), DAPI (blue); scale bar 20 μm. Mean +/- S.E.M shown, data representative of three independent experiments performed in triplicate. Statistical significance determined using ANOVA, with p values equaling *p<0.05, **p<0.01, ***p<0.005, ****p<0.001.

determined by densiometric evaluation of Western Blot bands (Fig 3J). Representative images of the immunofluorescence staining for PGC-1α (Fig 3K) and TFAM (Fig 3L) are shown with quantification of relative intensity (Fig 3M and 3N, respectively). These data suggests that Metformin can be used as to restore the attenuated expression of PGC-1α in NTM infected macrophages.

## Metformin and ZLN005 facilitate increased phagocytosis and intracellular killing of MAB

Macrophages target pathogens for destruction through phagocytosis and subsequent lysosomal degradation [42, 43]. We hypothesized that by increasing PGC-1α activation and rescuing mitochondrial biogenesis, macrophage phagocytic function would be enhanced. Therefore, we determined the effect of metformin and ZLN005 on phagocytosis. Macrophages that were treated with ZLN005 or Metformin were infected with fluorescent, labelled NTM and intracellular fluorescence was measured. Both metformin and ZLN005 treatment resulted in a significant increase in uptake of fluorescent particles relative to controls (Fig 4A). While MAB can form large, extracellular aggregates that prevent engulfment by macrophages, phagocytized MAB possesses the ability to avoid and prevent degradation [11, 44]. To evaluate the

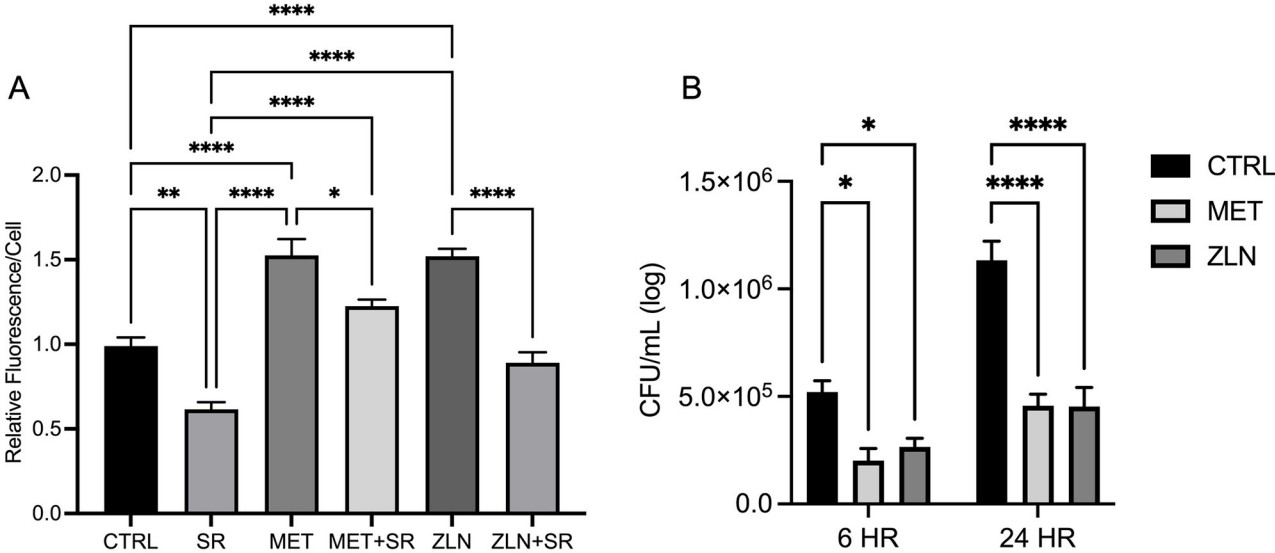

**Fig 4. Metformin and ZLN005 facilitate increased phagocytic uptake and intracellular killing of MAB.** Quantitation of phagocytic uptake of fluorescent *E. coli* in RAW 264.7 macrophages (A). Intracellular killing of MAB measured by CFU/mL(log) (B). Mean +/- S.E.M shown, data representative of three independent experiments performed in triplicate. Statistical significance determined using ANOVA, with p values equaling *p<0.05, **p<0.01, ***p<0.005.

effect of metformin and ZLN005 on macrophage-mediated destruction of MAB, drug-treated cells were infected and transferred to agar plates for CFU counting. Compared to controls, metformin and ZLN005 treatment elevated intracellular killing of MAB. We observed significantly decreased CFU numbers following both 6-hour infections and 24-hour infections (Fig 4B) irrespective of bacterial dilution. Overall, both metformin and ZLN005 exhibited comparable bactericidal activity, with metformin exhibiting a significant decrease in colony numbers compared to ZLN005 only in the 6-hour infection at a bacterial dilution of $10^{-7}$. Together these data for the first time show that Metformin and ZLN 005 enhance macrophage phagocytic and killing capacity of NTMs and hence have the potential to be used as immuno-modulators for refractory infections.

## Metformin and ZLN005 attenuate NTM mediated mitochondrial membrane potential damage

Our data shows that phagocytic capacity of macrophages is enhanced by metformin and ZLN005 treatment. We hypothesize that this is related to enhanced mitochondrial function of macrophages by increase in PGC-1α expression. NTM infection can induce mitochondrial dysfunction by disrupting the integrity of the inner mitochondrial membrane, causing a loss of mitochondrial membrane potential (MMP) [23, 28, 31]. To determine the influence of PGC-1α levels on mitochondrial membrane potential, human THP-1 macrophages were treated with metformin or ZLN005, PGC-1α inhibitor SR-18292 (SR), or a combination in cells infected with MAC. MMP integrity was evaluated through immunocytochemistry by measuring fluorescence expression of MitoTracker Green (MTG) and Tetramethylrhodamine methyl ester (TMRM). MAC infection significantly reduced TMRM fluorescence, indicative of damaged mitochondrial membrane potential, and this effect was exacerbated by treatment with SR. Metformin (Fig 5A and 5C) and ZLN005 (Fig 5B and 5D) treatment increased TMRM fluorescence compared to the control. Cells treated with metformin and ZLN005 and

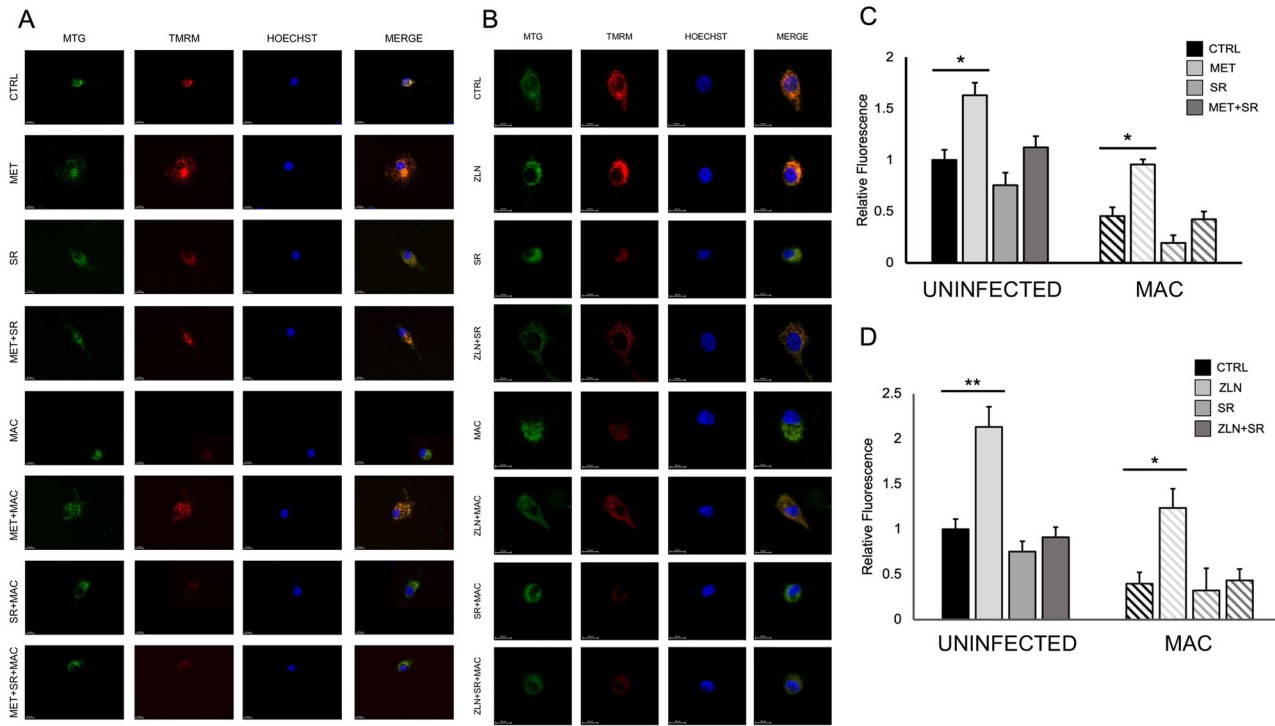

**Fig 5. Metformin and ZLN005 attenuate MAC-mediated mitochondrial membrane potential damage.** Representative confocal images of THP-1 macrophages treated with metformin and/or SR and infected with MAC (MOI 10) for 24 hours (A), and quantification of fluorescence expression (C). Representative confocal images of THP-1 macrophages treated with ZLN005 and/or SR and infected with MAC (MOI 10) for 24 hours (B), and quantification of fluorescence expression (D). MitoTracker Green (MTG) (green), TMRM (red), Hoechst (blue); scale bar 20 μm. Results determined through average fluorescence expression of TMRM normalized to MTG expression per DAPI-positive cell. Mean +/- S.E.M shown, data representative of three independent experiments performed in triplicate. Statistical significance determined using ANOVA, with p values equaling *p<0.05, **p<0.01.

infected with MAC exhibited elevated TMRM fluorescence compared to the untreated infected cells.

The level of TMRM fluorescence is an indicator of an intact and functional mitochondrial membrane. Together, these data suggest that metformin rescues MMP and damage induced by NTM, thus beneficial effects of Metformin are related to restoration of mitochondrial function.

## Beneficial effects of metformin are mediated through PGC-1α

Metformin is an FDA approved medication that is widely used for treatment of type II Diabetes Mellitus [40, 41, 45]. Previous studies in other models have established that the mechanism of action of metformin includes activation of AMPK and PGC-1α, and enhancement of mitochondrial biogenesis with improved bioenergetics [40, 45–47]. Since our data shows that Metformin rescues the expression of PGC-1α and enhances macrophage immune function we hypothesize that the beneficial effects are through activation of PGC-1α. To conclusively define if the beneficial effects are indeed through activation of PGC-1α we performed experiments where we treated cells with metformin and used PGC-1α inhibitor SR [48–51]. THP-1 macrophages were treated with metformin, SR, or a combination, and PGC-1α and TFAM protein expression was evaluated through immunocytochemistry (Fig 6A). In agreement with our previous results, macrophages treated with metformin exhibited significantly elevated protein expression of PGC-1α (Fig 6B) and TFAM (Fig 6C) compared to untreated controls. SR

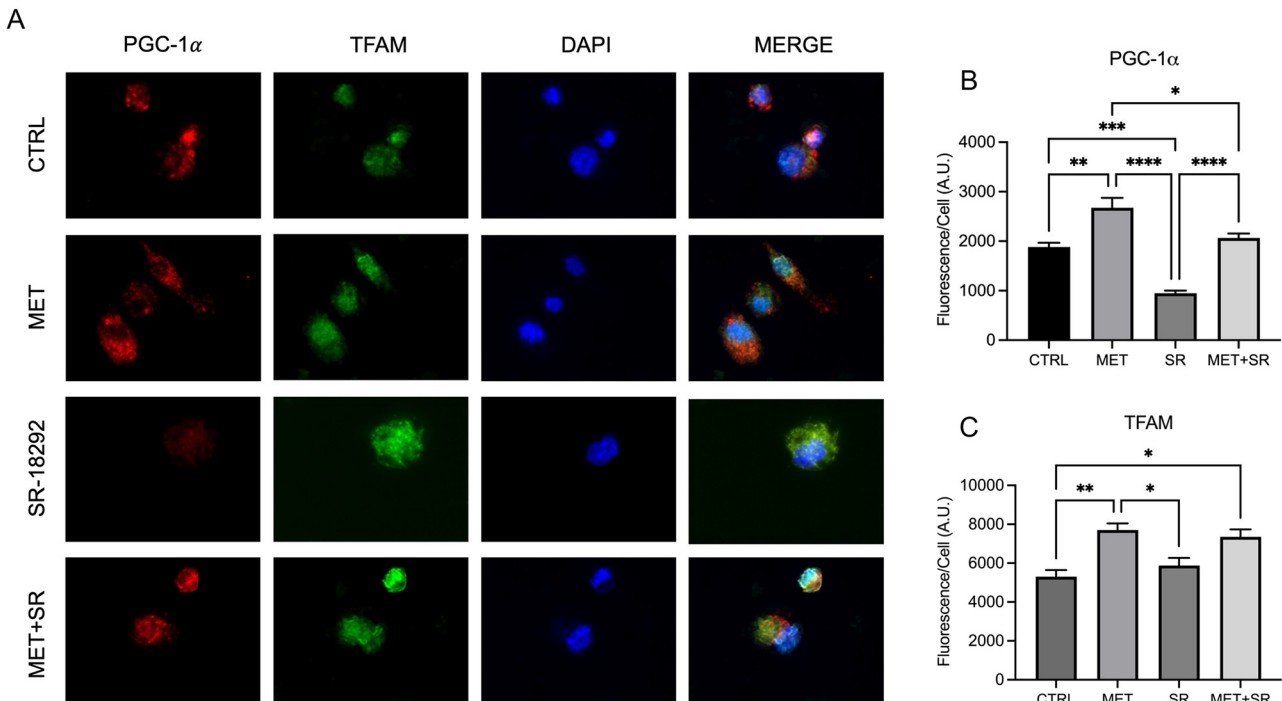

**Fig 6. Metformin rescues PGC-1α expression in presence of inhibitor.** Representative confocal images (A) of THP-1 macrophages treated with metformin and SR, and fluorescence quantitation of TFAM (B) and PGC-1α (C) protein expression. PGC-1α (red), TFAM (green), DAPI (blue); scale bar 20 μm. Mean +/- S.E.M shown, data representative of three independent experiments performed in triplicate. Statistical significance determined using ANOVA, with p values equaling *p<0.05, **p<0.01, ***p<0.005.

treatment had no measurable effect on TFAM expression, but significantly decreased PGC-1α expression compared to untreated controls. This decrease in PGC-1α fluorescence was abated in macrophages also treated with metformin. These data suggests that Metformin effects are likely mediated through PGC-1α.

## Discussion

NTM are gram-positive, acid-fast, aerobic bacilli that are ubiquitous in the environment and are important emerging pathogens [8, 9, 17]. Recent data suggest that NTM infections have surpassed tuberculosis in Western countries, yet there is a lack of understanding of host–pathogen interactions in NTM infections [18, 52]. Since mitochondria are the key powerhouse of the cell, affecting the function of immune cells, we investigated the impact of NTM on macrophage mitochondria. Our data show that macrophages infected with NTM demonstrate mitochondrial damage with attenuated phagocytosis. Mechanistically we show that NTM attenuates expression of PGC-1α, a key transcriptional regulator of mitochondrial biogenesis. During cellular stress new mitochondrial genesis is needed to rescue ATP production for cells to effectively execute key functions, in case of macrophage it is phagocytosis and bacterial killing [23, 34, 53]. We show that activation of PGC-1α enhances phagocytosis and bactericidal activity in NTM infected macrophages.

The immune response generated against NTM infections encompasses multiple cell types, however macrophages are the primary cells involved in initial defense against inhaled mycobacteria, making their activity indispensable for an efficient immune response [6, 16, 17, 19]. Efficient and effective macrophage function against NTMs and other pathogens is highly

dependent upon cellular energetics and energy production [26, 28, 54]. Preventing damage to immune cell mitochondria is paramount in defense against bacterial infections [55, 56]. Mitochondria play integral roles in immune cell function, not only in energy production but also in producing ROS used for bactericidal activity [16, 28, 57].

NTMs have evolved multiple mechanisms to prevent macrophage-mediated phagocytic uptake and degradation, through disrupting mitochondrial energetics of immune cells [8, 9]. MAB and MAC accomplish this through inducing production of mtROS and decreasing the expression of antioxidant molecules involved in neutralizing inflammatory molecules [28]. NTM infections can lead to excessive and damaging ROS production that causes cellular damage and mitochondria dysfunction culminating in apoptotic, necrotic, or pyroptotic cell death [8, 58]. This can induce the production and release of miRNA, lipid mediators, cellular fragments, cytokines, and chemokines that trigger activation of damage associated molecular patterns (DAMPs) and inflammatory signaling pathways; this cycle of bacteria-mediated damage and mitochondrial dysfunction further drives the inflammatory environment in lung tissue, preventing effective host immune response [22, 28, 59, 60].

Regulating mitochondrial function is of paramount importance for maintaining proper cellular function of immune cell and macrophage metabolism, as the balance of mitochondrial biogenesis and mitophagy regulates the number and processes of mitochondria [23, 28, 61, 62]. Excessive mitochondrial damage can result in dysfunctional metabolism, and within the framework of bacterial infection, dysfunctional mitochondria function can disrupt immune cell function and result in a deficit in immune response [23, 24, 28, 50, 55, 56, 63, 64]. We investigated macrophage mitochondria function through multiple mechanisms, including evaluating mtDNA damage and inner mitochondrial membrane integrity. mtDNA damage was determined by evaluating the 16S-RNA gene copy numbers of short mtDNA-79 fragments and full-length mtDNA-230, with the 79:230 ratio used to evaluate mitochondrial genomic integrity [65]. In agreement with this, our research found that MAB and MAC infection caused a significant increase in the 79:230 ratio, indicative of mitochondrial damage in infected macrophages. This was further supported by the significant increase in mitochondrial superoxide accumulated in macrophages infected with MAC compared to uninfected controls. In accordance with other literature, MAB and MAC induce mitochondrial damage in macrophages that result in induced deficits in integrity of both mtDNA and MMP [22, 66]. The combined data of decreased mitochondria number and decreased mtDNA integrity show that NTM infection can drastically alter host cell energy production through damage to mitochondria. Deficits in energy production and mitochondrial activity negatively impact the ability of macrophages and other immune cells to effectively combat bacterial infections [23, 28].

To determine the mechanisms by which mitochondrial function is deranged, we investigated the impact of NTM on key transcriptional regulators of mitochondrial biogenesis. PGC-1α expression induces biogenesis of mitochondria through autophagy and processing damaged mitochondria and production of healthy mitochondria through either fission or fusion reactions [25, 26, 31, 62]. PGC-1α is the master regulator of mitochondrial biogenesis, and controls production of new mitochondria through either fission or fusion [6, 26, 27, 62, 67]. Immune cells with PGC-1α downregulated possess dysfunctional metabolic processes, resulting in a reduced capacity to clear infections [7, 19, 39, 55, 68]. In addition to control over mitochondrial biogenesis, PGC-1α exerts multiple beneficial outcomes in immune cells, including expression of cytoprotective and antioxidant proteins. Inflammatory signaling induced by the presence of NTM results in excessive ROS production which can damage mitochondria and reduce function [6, 28, 69]. Induction of PGC-1α expression attenuates excessive inflammatory signaling, reducing macrophages and immune cells polarization and differentiation into pro-inflammatory cells [27, 28, 31, 70–72].

TFAM—which is regulated by PGC-1α—also influences mitochondrial function by regulating mtDNA replication and gene transcription, and stability and genomic integrity of mtDNA [54, 73]. Similar to PGC-1α, expression of TFAM is critical for proper mitochondrial activity and function, with cells lacking TFAM exhibiting morphological damage to mitochondria, loss of mtDNA integrity and copy number, and deficiencies in oxidative phosphorylation and energy production [54, 63, 73]. Alveolar macrophages from mice with TFAM mutations possessed reduced numbers of mitochondria with abnormal morphology, decreased oxygen consumption rate and maximal mitochondrial respiratory capacity [73]. In addition to metabolic and energetic deficits, the loss of TFAM expression also impaired self-renewal and proliferation of macrophages [54, 73]. TFAM deficiency also corresponded to an increase in inflammatory cytokine production and a reduction in antioxidant gene expression, both of which contribute to mtDNA damage [64, 73]. Our data show that NTM infections *in vitro* downregulate both PGC-1α and TFAM, two critical determinants of mitochondrial function that can impact the function and immune status of macrophages.

ROS production is necessary for induction of signaling pathways and are also generated by immune cells to kill bacteria, however excessive ROS production activates inflammatory signaling pathways that disrupt normal cellular activities [6, 19, 23, 59]. PGC-1α also promotes and regulates antioxidant gene expression in response to elevated ROS production [6, 74]. Balancing the production of ROS is tantamount to proper metabolic functions, and increasing PGC-1α expression—through pharmaceutical agents or genetic modification—has beneficial effects on host cell energetics and can reduce the harmful effects of elevated ROS and inflammatory cytokine production [26, 31, 69, 75, 76]. While it is necessary for macrophages to produce ROS for bactericidal activity against phagocytized bacteria, excessive concentrations of intracellular ROS damage mitochondria and diminish their energetic capacity [16, 23, 28, 72]. This can result in reduced immune activity, preventing effective clearance of bacteria and allowing the infection to progress [6, 24, 77]. Thus, inducing PGC-1α expression in macrophages aids in defense against infection by regulating mitochondrial activity and the production of anti-inflammatory enzymes, making PGC-1α an attractive cellular target for host-directed therapy against NTM infections [26, 67, 78].

Next, we determined if rescuing mitochondrial biogenesis by inducing PGC-1α can restore macrophage immune response. Approaches to activate PGC-1α include small molecule activators such as ZLN005, which transcriptionally activates PGC-1α and metformin which is a widely used anti-diabetic agent [31, 40, 41, 79]. Metformin has been shown to reduce the severity of bacterial infections in preclinical and clinical retrospective studies, although the exact mechanisms remain elusive [80]. The potential mechanisms by which Metformin exerts beneficial effects and rescues mitochondrial bioenergetics are related to its activation of AMPK and downstream activation of PGC-1α [39]. Therefore, we tested metformin as a PGC-1α activator in our *in vitro* model. Cells that were treated with ZLN005 and metformin showed an increased activation of PGC-1α, TFAM with increased phagocytosis and bacterial killing. The beneficial effects of metformin were blunted when cells were treated with siPGC-1α or inhibitors of PGC-1α suggesting that the effects of metformin were indeed mediated through PGC-1α. (Fig 7).

Traditional antibiotics have become increasingly unreliable in treating bacterial infections, with emerging drug-resistant strains that are difficult to treat, thus focusing on host-directed therapies can avoid the issues of antibiotic resistance through enhancement of immune cell function [1, 6, 8]. Preventing metabolic dysfunction in immune cells is paramount to mounting an effective and efficient response to bacterial infection [58, 81, 82]. Bacterial infection can influence metabolic function, causing morphological damage to mitochondria that can reduce energetic output and ultimately prevent proper activity of macrophages and other immune

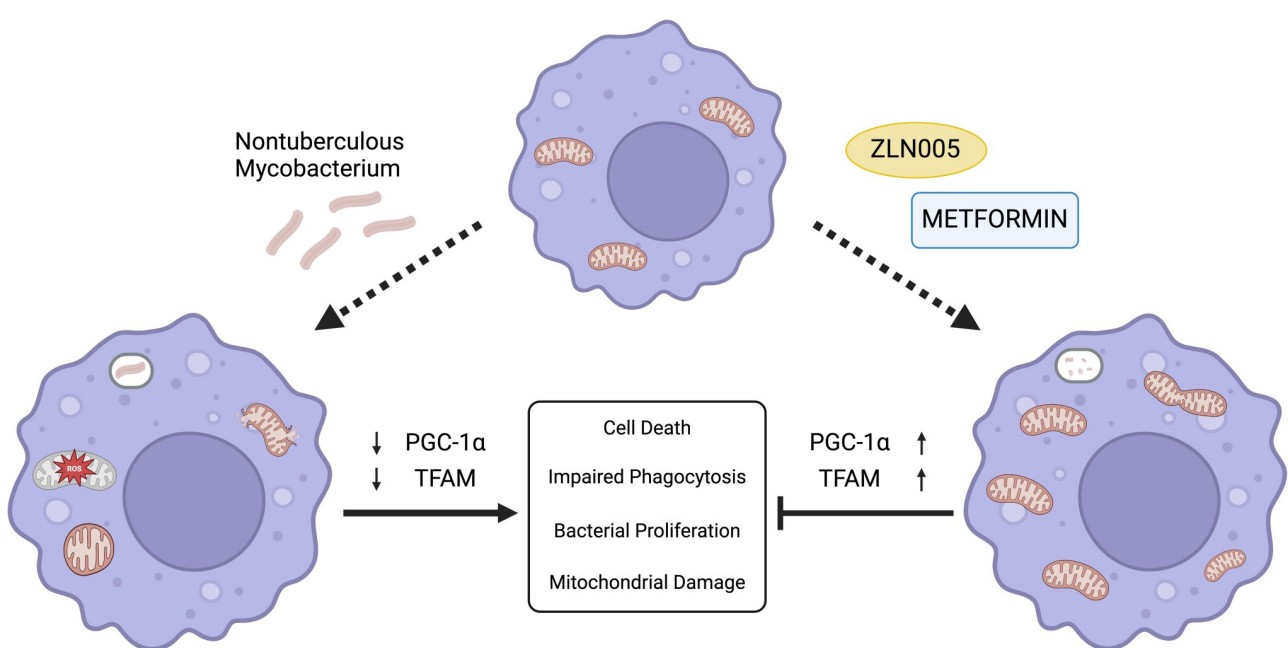

**Fig 7. Metformin and ZLN005 prevent NTM-mediated macrophage dysfunction through PGC-1α.** Macrophages infected with NTMs exhibit decreased PGC-1α and TFAM expression, leading to mitochondrial dysfunction and reduced antibacterial activity. Treatment with metformin and ZLN005 were shown to induce PGC-1α and TFAM expression, and enhance the macrophage-mediated response against NTM, preventing mitochondrial dysfunction, bacterial proliferation, and cell death.

cells [22, 31, 39, 55, 78]. Our data show that rescue of mitochondrial biogenesis through activation of PGC-1α can rescue the macrophage immune function and enhance clearance of NTM.

In conclusion, our study shows that NTM infected macrophages exhibit mitochondrial damage with impaired bioenergetics and increased ROS production. Mechanistically, we show that impaired mitochondrial function is a result of attenuated expression of PGC-1α and TFAM, key transcriptional activators of mitochondrial biogenesis. Impaired mitochondrial function compromises macrophage ability to phagocytose and kill NTM. Most importantly activating PGC-1α by utilizing small molecule activators or metformin restored mitochondrial function and macrophage immune response. These data suggest that PGC-1α activation—if combined with antibiotic treatment—will provide novel therapeutic approach to enhance host defenses against NTM infections. Future translational studies are needed to define immuno-modulatory approaches for NTM infections which is an unmet medical need.

## Supporting information

**S1 Raw images. Full Western blot images from manuscript.** Raw Western blot images from the manuscript. Blots were incubated with primary antibodies and HRP-conjugated secondary antibodies and acquired through chemiluminescence imaging. Fig 2: PGC-1α (A, D), TFAM (B, E), β-Actin (C, F); Fig 3: PGC-1α (G, J), TFAM (H, K), β-Actin (I, L).
(TIF)

**S1 Fig. ZLN005 and metformin cell toxicity.** THP-1 macrophages were treated with a range of concentrations of metformin (A) and ZLN005 (B) for 24 hours. An MTT assay was utilized to determine toxicity and the optimal drug concentration for treatment of cells.
(TIF)

**S2 Fig. ZLN005 and metformin toxicity in MAC.** MAC cultures were treated with a range of concentrations of metformin (A) and ZLN005 (B) for 24 hours, and the $OD_{600}$ was measured to determine toxicity to MAC. Concentrations used in cell culture treatments was found to have no significant influence on MAC *in vitro*.
(TIF)

## Author Contributions

**Conceptualization:** Ruxana T. Sadikot.

**Data curation:** Joel R. Frandsen, Zhihong Yuan, Brahmchetna Bedi, Zohra Prasla, Seoung-Ryoung Choi, Prabagaran Narayanasamy.

**Funding acquisition:** Ruxana T. Sadikot.

**Investigation:** Joel R. Frandsen, Zhihong Yuan, Brahmchetna Bedi, Zohra Prasla, Seoung-Ryoung Choi, Prabagaran Narayanasamy.

**Methodology:** Zhihong Yuan, Prabagaran Narayanasamy.

**Supervision:** Ruxana T. Sadikot.

**Writing – original draft:** Joel R. Frandsen.

**Writing – review & editing:** Ruxana T. Sadikot.

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
