## [Decision Letter · Decision Letter 0]

5 Dec 2023

PONE-D-23-34313PGC-1α Activation to Enhance Macrophage Immune Function in Mycobacterial InfectionsPLOS ONE

Dear Dr. Sadikot,

Thank you for submitting your manuscript to PLOS ONE. After careful consideration, we feel that it has merit but does not fully meet PLOS ONE’s publication criteria as it currently stands. Therefore, we invite you to submit a revised version of the manuscript that addresses the points raised during the review process.

Please submit your revised manuscript by Jan 19 2024 11:59PM. If you will need significantly more time to complete your revisions, please reply to this message or contact the journal office at plosone@plos.org. Please include the following items when submitting your revised manuscript:A rebuttal letter that responds to each point raised by the academic editor and reviewer(s). You should upload this letter as a separate file labeled 'Response to Reviewers'.A marked-up copy of your manuscript that highlights changes made to the original version. You should upload this as a separate file labeled 'Revised Manuscript with Track Changes'.An unmarked version of your revised paper without tracked changes. You should upload this as a separate file labeled 'Manuscript'.

We look forward to receiving your revised manuscript.

Kind regards,

Frederick Quinn

Academic Editor

PLOS ONE

Journal Requirements:

"This research was funded through grants from the U.S. Department of Veterans Affairs

 (BX001786), and National Institutes of Health/The National Heart, Lung, and Blood Institute (R01-HL144478)"

Reviewers' comments:

Reviewer's Responses to Questions

**Comments to the Author**

1. Is the manuscript technically sound, and do the data support the conclusions?

Reviewer #1: Partly

Reviewer #2: Yes

2. Has the statistical analysis been performed appropriately and rigorously? 

Reviewer #1: Yes

Reviewer #2: Yes

3. Have the authors made all data underlying the findings in their manuscript fully available?

Reviewer #1: Yes

Reviewer #2: Yes

4. Is the manuscript presented in an intelligible fashion and written in standard English?

Reviewer #1: Yes

Reviewer #2: Yes

5. Review Comments to the Author

Reviewer #1: This manuscript, titled 'PGC-1α Activation to Enhance Macrophage Immune Function in Mycobacterial Infections,' aimed to investigate the impact of NTM infections on host macrophage mitochondria. Importantly, NTM disrupted the ability of macrophages to generate functional mitochondria by attenuating PGC-1α and TFAM, which were subsequently restored by PGC-1α pharmacologic activators. However, the in vivo studies lack solid evidence to conclusively prove that the altered expression of PGC-1α or TFAM, caused by NTM, occurred in macrophages. Furthermore, the fluorescence images are obscure, as evidenced in Fig. 1C, Fig. 3K, Fig. 4K, and Fig. 4L.

Reviewer #2: This is a nice manuscript and worthy of publication. However, appreciate author’s comments/responses to the following points:

1. In general, language requires some revision and editing particularly in the “Discussion” section which requires significant revision for better clarity and removal of redundant and repetitive sentences.

2. Line 153: What dose of oral metformin in an average human would be equivalent to the final concentration of 2 mM metformin mentioned in the manuscript. Please add to the manuscript.

3. Line 182: Is the correct number 15,000 RPM and not 150,000?

4. Lines 245, 293, 303: Please specify what the cells are treated with

5. Figure 1C and 1D:

a. The MitoSOX staining for CTRL seems to be nothing (at least not visible on the provided image) where as mentioned as about 1 on Figure 1D. Appreciated comment clarification.

b. MitoSox-based flow cytometry could be considered as an alternative

6. Figure 2: Whole lung tissue was used for this experiment and thus it is not clear if the results seen are due to changes in macrophages, other cells, or both. This is important as the authors have previously shown that PGC-1a overexpression, resveratrol, and metformin enhance host bronchial epithelial cell mitochondrial function and improve epithelial innate response to P. aeruginosa. Showing the changes specifically in macrophages (by cell sorting or staining) will be important as they are the focus of the manuscript.

7. Figure 3 related:

a. There is an issue with discrepancy between the text (line 391 to 403) and figures 3A to 3H where the text states “MAB-mediated” but figures show “MAC”.

b. However, Figures 3I and 3J mention MAB.

c. Please reconcile.

d. Figure 3K states MAB but 3M mentions MAC. Please clarify

e. Also, the morphology of the cells in the bottom row of figure 3K appear different from the rest. In our experience, the morphology of cells (roundish) in the 3 top rows is more in line with monocytes while the morphology in the bottom row is more in line with activated macrophages. Appreciate authors’ input.

f. Pre-treatment with ZLN005 was used in this experiment, would treatment after MAB or MAC infection produce similar results and be as effective? This is of clinical importance as treatment is usually initiated after the infection.

8. Figure 4 related:

a. Would treatment with metformin after the infection produce the similar results?

9. Figure 5:

a. SR (I assume stands for SR-18292) is mentioned in the figure but no explanation in the text or figure legend.

10. Line 492: Please clear the track changes (please remove “and macrophages” at the end of the sentence.

11. Lines 521-523: Please revise the sentence for better clarity

12. Lines 532-534: Given that TFAM is downstream from PGC-1a (as also mentioned in the manuscript) why SR-18292 had no measurable effect on TFAM despite significantly reducing PGC-1a expression?

13. Line 596: Please add “(mitochondrial membrane potential)” after ΔΨm.

14. A figure depicting the pathways and findings will be a welcomed addition

Thank you

6. PLOS authors have the option to publish the peer review history of their article (what does this mean?). If published, this will include your full peer review and any attached files.

Reviewer #1: No

Reviewer #2: No

---

## [Author Response · Author response to Decision Letter 0]

5 Feb 2024

Reviewer #1: 

1) Comment: This manuscript, titled 'PGC-1α Activation to Enhance Macrophage Immune Function in Mycobacterial Infections,' aimed to investigate the impact of NTM infections on host macrophage mitochondria. Importantly, NTM disrupted the ability of macrophages to generate functional mitochondria by attenuating PGC-1α and TFAM, which were subsequently restored by PGC-1α pharmacologic activators. However, the in vivo studies lack solid evidence to conclusively prove that the altered expression of PGC-1α or TFAM, caused by NTM, occurred in macrophages. 

Response: We appreciate the reviewer comment about the significance of our work, we agree that the in vivo model does not necessarily reflect the expression of the mitochondrial proteins only in the macrophages. They are probably a combination of multiple cell types. The purpose of the in vivo experiment is to demonstrate that the expression of the mitochondrial proteins is altered in the lung. We do agree that more in depth studies are needed to define the in vivo role of macrophages in the lungs. In future studies, we plan to study the in vivo model in depth, however, it is beyond the scope of the current manuscript. 

2) Comment: The fluorescence images are obscure, as evidenced in Fig. 1C, Fig. 3K, Fig. 4K, and Fig. 4L.

Response: We have included labeling with the colors that are represented by the fluorescent images. This provides the explanation for the images; this is now included in the results and figure legends. 

Reviewer #2: 

Comment: This is a nice manuscript and worthy of publication. However, appreciate author’s comments/responses to the following points. We appreciate the reviewer comments and have addressed the concerns raised. 

1. Comment: In general, language requires some revision and editing particularly in the “Discussion” section which requires significant revision for better clarity and removal of redundant and repetitive sentences.

Response: We appreciate the comment and have now carefully edited the manuscript for language and revised the discussion section for brevity and clarity (Lines 570-705)

2. Comment: Line 153: What dose of oral metformin in an average human would be equivalent to the final concentration of 2 mM metformin mentioned in the manuscript. Please add to the manuscript.

Response: This is an important point, thank you for raising this issue. Metformin dosages prescribed for diabetes range up to a maximum of 2.5 g (35 mg/kg) per day, which corresponds to blood sera levels of 10-20 µM. Pharmacokinetic evidence shows metformin’s action in the mitochondria occurs through a gradual build-up in the inner mitochondrial matrix, which can have concentrations orders of magnitude higher than serum levels. In one study, a serum concentration of 10 µM was correlated to mitochondrial concentrations 100 times higher. Thus, the elevated metformin concentrations used in this study and others correlates to intramolecular concentration of metformin in vivo.

3. Comment: Line 182: Is the correct number 15,000 RPM and not 150,000?

Response: Correct, the centrifugation speed is “15,000 RPM” (Line 182), this has been corrected

4. Comment: Lines 245, 293, 303: Please specify what the cells are treated with?

Response: We have now included specifics about drug treatments and concentrations. Cells were treated with metformin [2 mM], ZLN005 [2 µM], and SR-18292 [10 µM] (Line 245, 293, 305), 

5. Comment: Figure 1C and 1D:

a. The MitoSOX staining for CTRL seems to be nothing (at least not visible on the provided image) where as mentioned as about 1C on Figure 1D.

Response: This figure has now been replaced. 

b. MitoSox-based flow cytometry could be considered as an alternative

Response: We agree that flow cytometry can be alternative, however, the reason for using fluorescent cell staining was to illustrate and visualize the progressive increase in MitoSOX fluorescence in macrophages during the initial stages of bacterial infection.

6. Comment: Figure 2: Whole lung tissue was used for this experiment and thus it is not clear if the results seen are due to changes in macrophages, other cells, or both. This is important as the authors have previously shown that PGC-1a overexpression, resveratrol, and metformin enhance host bronchial epithelial cell mitochondrial function and improve epithelial innate response to P. aeruginosa. Showing the changes specifically in macrophages (by cell sorting or staining) will be important as they are the focus of the manuscript.

Response: We agree with the reviewer the in vivo experiments do not discriminate between the cell types are demonstrating the upregulation. We agree that the in vivo model does not necessarily reflect the expression of the mitochondrial proteins only in the macrophages. They are probably a combination of multiple cell types. The purpose of the in vivo experiment is to demonstrate that the expression of the mitochondrial proteins is altered in the whole lung. We do agree that more in depth studies are needed to define the in vivo role of macrophages in the lungs. In future studies, we plan to study the in vivo model in depth, however, it is beyond the scope of the current manuscript since the focus of the current manuscript is to investigate the effects on macrophages in vitro. While we cannot be certain the effect seen was due to cells other than alveolar macrophages, the in vivo experiments demonstrate changes in the whole lung tissue. We agree that in future experiments, it would be beneficial to isolate macrophages from the lung tissue to determine their specific influence on PGC-1α expression during NTM infection. 

7. Comment: Figure 3 related:

a. There is an issue with discrepancy between the text (line 391 to 403) and figures 3A to 3H where the text states “MAB-mediated” but figures show “MAC”. b. However, Figures 3I and 3J, 3K mention MAB. Please reconcile.

7a-d. Response: We regret the oversight in labeling, figures and captions, are all revised. 

e. Also, the morphology of the cells in the bottom row of figure 3K appear different from the rest. In our experience, the morphology of cells (roundish) in the 3 top rows is more in line with monocytes while the morphology in the bottom row is more in line with activated macrophages. Appreciate authors’ input.

Response: We appreciate the comment, the immunocytochemistry images are that of macrophages, which exhibit both circular and irregular/stellate shaped morphology.

Comment: Pre-treatment with ZLN005 was used in this experiment, would treatment after MAB or MAC infection produce similar results and be as effective? This is of clinical importance as treatment is usually initiated after the infection.

Response: We agree that treatment post infection would have translational implications, these experiments need careful dose response and time course experiments, are ongoing and will be tested in the future. 

8. Comment: Figure 4 related: Would treatment with metformin after the infection produce the similar results?

Response: We anticipate similar results with metformin post treatment, since it is a potent PGC-1α activator. 

9. Comment: Figure 5: . SR (I assume stands for SR-18292) is mentioned in the figure but no explanation in the text or figure legend.

Response: Correct, we have included this in the figure caption SR-18292 and in the abbreviation SR. (Line 520)

10. Comment: Line 492: Please clear the track changes (please remove “and macrophages” at the end of the sentence.

Response: Thank you, we have corrected this. “and macrophages” was deleted from sentence. (Line 492).

11. Comment: Lines 521-523: Please revise the sentence for better clarity

Response: We have revised lines 521-523 and made corrections for better clarity.

12. Comment: Lines 532-534: Given that TFAM is downstream from PGC-1a (as also mentioned in the manuscript) why SR-18292 had no measurable effect on TFAM despite significantly reducing PGC-1a expression?

Response: There are multiple signaling pathways and transcription factors involved with the activation and regulation of PGC-1α and TFAM, and both can be activated through PPARγ signaling. Thus there may be other potential pathways for activation of TFAM, additionally, post transcriptional modifications may be involved in differential gene expression between PGC-1α and TFAM.

13. Comment: Line 596: Please add “(mitochondrial membrane potential)” after ΔΨm.

Response: Thank you, this has been corrected.

14. Comment: A figure depicting the pathways and findings will be a welcomed addition

Response: We appreciate the suggestion and have added a figure (Figure 8).

---

## [Decision Letter · Decision Letter 1]

19 Mar 2024

PONE-D-23-34313R1PGC-1α Activation to Enhance Macrophage Immune Function in Mycobacterial InfectionsPLOS ONE

Dear Dr. Sadikot,

Thank you for submitting your manuscript to PLOS ONE. After careful consideration, we feel that it has merit but does not fully meet PLOS ONE’s publication criteria as it currently stands. Therefore, we invite you to submit a revised version of the manuscript that addresses the points raised during the review process, particularly major concern from Reviewer 1. Please submit your revised manuscript by May 03 2024 11:59PM. If you will need significantly more time to complete your revisions, please reply to this message or contact the journal office at plosone@plos.org. Please include the following items when submitting your revised manuscript:A rebuttal letter that responds to each point raised by the academic editor and reviewer(s). You should upload this letter as a separate file labeled 'Response to Reviewers'.A marked-up copy of your manuscript that highlights changes made to the original version. You should upload this as a separate file labeled 'Revised Manuscript with Track Changes'.An unmarked version of your revised paper without tracked changes. You should upload this as a separate file labeled 'Manuscript'.

We look forward to receiving your revised manuscript.

Kind regards,

Frederick Quinn

Academic Editor

PLOS ONE

Reviewers' comments:

Reviewer's Responses to Questions

**Comments to the Author**

1. If the authors have adequately addressed your comments raised in a previous round of review and you feel that this manuscript is now acceptable for publication, you may indicate that here to bypass the “Comments to the Author” section, enter your conflict of interest statement in the “Confidential to Editor” section, and submit your "Accept" recommendation.

Reviewer #1: (No Response)

Reviewer #2: (No Response)

2. Is the manuscript technically sound, and do the data support the conclusions?

Reviewer #1: Partly

Reviewer #2: Yes

3. Has the statistical analysis been performed appropriately and rigorously? 

Reviewer #1: Yes

Reviewer #2: N/A

4. Have the authors made all data underlying the findings in their manuscript fully available?

Reviewer #1: Yes

Reviewer #2: Yes

5. Is the manuscript presented in an intelligible fashion and written in standard English?

Reviewer #1: Yes

Reviewer #2: No

6. Review Comments to the Author

Reviewer #1: The revised manuscript has been greatly improved. However, the in vivo studies can not prove that the altered expression of PGC-1α or TFAM, caused by NTM, occurred in macrophages.

Reviewer #2: I thank the authors for their responses.

1. Line 152 (revised version): What dose of oral metformin in an average human would be equivalent to the final concentration of 2 mM metformin mentioned in the manuscript. Please add to the manuscript. Appreciate authors’ comments. It would be a plus for the manuscript if their explanation could be incorporated into the manuscript.

2. Line 305 (revised version): Please add what the cells are treated with.

3. Figure 1C and 1D: I think there should be a mistake as the picture appears to be the same and not replaced (please see below).

a. The MitoSOX staining for CTRL seems to be nothing (at least not visible on the provided image) where as mentioned as about 1 on Figure 1D. Appreciated comment clarification.

b. MitoSox-based flow cytometry could be considered as an alternative.

4. Figure 2: I appreciated author’s comments. As mentioned in the original comments showing the changes specifically in macrophages (by cell sorting or staining) is very important given the focus of this manuscript and the author’s prior work. These experiments should be very straightforward and will significantly improve the manuscript. However, if further experiments are not feasible or can’t be done then this section could be omitted particularly given that the per authors the in vivo studies are beyond the scope of this manuscript or at least the limitation clearly mentioned.

5. Lines 434-436: Language needs revision for better clarity.

6. Figure 4 related:

a. Text mentions PGC-1a while figure A and B mention “ppargc1a” (e.g., line 440 could change to “decreases in PGC1-a mRNA (ppargc1a)”.

b. Figure 4 (K, L, M, N): While figures K and L mention “ZLN + MAC”, figures M and N and the text mention MET + MAC. Which one is it? The correct image should be MET + MAC to confirm to the text and the rest of images. Please rectify.

7. Figure 5: Again:

a. SR (I assume stands for SR-18292) is mentioned in the figure but no mention or explanation of SR in either the text or the figure legend.

b. Although SR is mentioned on line 510 (and not 520) and under next topic and in relation to Figure 6, it should be mentioned at the first appearance for clarity.

8. Figure 6:

a. A and B. Moving rows 7 to just above current row 4 (MAC) will help the clarity in relative to C and D.

b. Also, the legend incorporated in the figures C and D needs to be adjusted to incorporate explanation for the “infection” part of the images.

9. Line 541-542: Please consider changing “and enhances” to “and enhancement of”.

10. Figure 7:

a. Are graphs B and C based on Western Blot (not shown) or immunofluorescent images (A). If based on Western Blot then please add blot images. If based on immunofluorescent then the almost doubling of the expression noted in “B” (MET vs CTRL) does not appear to be supported by the image in “A”.

11. Lines 575 ot 577:

a. “Bactericidal activity” (used later in the text) could be considered instead of “bacterial killing”.

12. Line 579: Please consider “involved” instead of “functioning”.

13. Line 612: Please consider removing “our”.

14. Line 614: Please consider changing “resulting” to “results” and the “ΔΨm” to “MMP” as mentioned earlier in the manuscript.

15. Line 618: Please either consider removing “and immune cells” or adding “perhaps other” before it.

16. Lines 631-632: Please consider minor revision for clarity.

17. Lines 650-651: Please consider revision/breaking the sentence.

18. Lines 672-674: Please consider revision for better clarity.

19. Line 691: Please consider adding “and” after “bioenergetics,”.

7. PLOS authors have the option to publish the peer review history of their article (what does this mean?). If published, this will include your full peer review and any attached files.

Reviewer #1: No

Reviewer #2: No

---

## [Author Response · Author response to Decision Letter 1]

16 May 2024

Response to Reviewers: 

Reviewer #1: 

Comment: The revised manuscript has been greatly improved. However, the in vivo studies can not prove that the altered expression of PGC-1α or TFAM, caused by NTM, occurred in macrophages.

Response: We agree with the reviewer and have now removed the in vivo data and are working on more in details studies. This manuscript will focus on the in vitro mechanistic studies. 

Reviewer #2: I thank the authors for their responses.

Comment: Line 152 (revised version): What dose of oral metformin in an average human would be equivalent to the final concentration of 2 mM metformin mentioned in the manuscript. Please add to the manuscript. Appreciate authors’ comments. It would be a plus for the manuscript if their explanation could be incorporated into the manuscript.

Response: Appreciate the reviewer bringing up the issue of dose of Metformin. The maximum dose of oral metformin in clinical practice is 2.5–3 g per day, which is approximately 35–42 mg/kg. In humans, the plasma concentration range of metformin after passing through the liver is usually 10–40 μM. For an invitro dose, 2 mM is would be comparable to the human doses on the higher side. This explanation has been added to the discussion of the manuscript. 

Comment: Line 305 (revised version): Please add what the cells are treated with.

Response: We have now added the cells. “THP-1 cells were cultured, treated with metformin [2 mM], ZLN005 [2 µM], and SR [10 µM], and infected in 96-well plates (Grenier Bio-One) as previously described.”

Comment: Figure 1C and 1D: I think there should be a mistake as the picture appears to be the same and not replaced (please see below).

a. The MitoSOX staining for CTRL seems to be nothing (at least not visible on the provided image) where as mentioned as about 1 on Figure 1D. Appreciated comment clarification.

b. MitoSox-based flow cytometry could be considered as an alternative.

Response: We appreciate the comment and regret the oversight. Figure 1C was updated to include a new image panel for the Control group, and 1D was revised with updated statistical analysis. The Control group exhibited low expression of MitoSOX staining in all replicates used for analysis. This was seen in multiple experiments performed.

Comment: Figure 2: I appreciated author’s comments. As mentioned in the original comments showing the changes specifically in macrophages (by cell sorting or staining) is very important given the focus of this manuscript and the author’s prior work. These experiments should be very straightforward and will significantly improve the manuscript. However, if further experiments are not feasible or can’t be done then this section could be omitted particularly given that the per authors the in vivo studies are beyond the scope of this manuscript or at least the limitation clearly mentioned.

Response: We appreciate the reviewer comments and have now opted to remove the in vivo data. 

Comment: Lines 434-436: Language needs revision for better clarity.

Response: This was related to the in vivo data, which has been removed, the paragraph has been revised accordingly. 

Comment: Figure 4 related:

a. Text mentions PGC-1a while figure A and B mention “ppargc1a” (e.g., line 440 could change to “decreases in PGC1-a mRNA (ppargc1a)”.

b. Figure 4 (K, L, M, N): While figures K and L mention “ZLN + MAC”, figures M and N and the text mention MET + MAC. Which one is it? The correct image should be MET + MAC to confirm to the text and the rest of images. Please rectify.

Response: 

a. The text was edited to specify the protein and RNA forms of PGC-1α and TFAM; PGC-1α mRNA as “ppargc1a” and TFAM mRNA as “tfam”

b. The figures are edited to add the proper treatment conditions.

Comment: Figure 5: Again:

a. SR (I assume stands for SR-18292) is mentioned in the figure but no mention or explanation of SR in either the text or the figure legend.

b. Although SR is mentioned on line 510 (and not 520) and under next topic and in relation to Figure 6, it should be mentioned at the first appearance for clarity.

Response: 

Appreciate the comment, regret the oversight. This has been corrected. SR-18292 is referred to as “SR” first in the materials and methods (line 154), and in the main text body (line 540)

Comment: Figure 6:

a. A and B. Moving rows 7 to just above current row 4 (MAC) will help the clarity in relative to C and D.

b. Also, the legend incorporated in the figures C and D needs to be adjusted to incorporate explanation for the “infection” part of the images.

Response: The image panels were adjusted for better clarity and organization. The figure legend was updated to include “Uninfected” and “MAC” on the X axis.

Comment: Line 541-542: Please consider changing “and enhances” to “and enhancement of”.

Response: This has been corrected, “and enhances” was revised to “and enhancement of”

Comment: Figure 7:

a. Are graphs B and C based on Western Blot (not shown) or immunofluorescent images (A). If based on Western Blot then please add blot images. If based on immunofluorescent then the almost doubling of the expression noted in “B” (MET vs CTRL) does not appear to be supported by the image in “A”.

Response: The graphs for Figure 7 are quantified from the captured immunofluorescent images. The data from the immunofluorescent images was re-quantified and reanalyzed to provide a revised graphical representation.

Comment: Lines 575 ot 577:

a. “Bactericidal activity” (used later in the text) could be considered instead of “bacterial killing”.

Response: This has been revised, we show that activation of PGC-1α enhances phagocytosis and bactericidal activity in NTM infected macrophages.

Comment: Line 579: Please consider “involved” instead of “functioning”.

Response: This is edited to, “the immune response generated against NTM infections encompasses multiple cell types, however macrophages are the primary cells involved in initial defense against inhaled mycobacteria, making their activity indispensable for an efficient immune response”

Comment: Line 612: Please consider removing “our”.

Response: “our” has been removed

Comment: Line 614: Please consider changing “resulting” to “results” and the “ΔΨm” to “MMP” as mentioned earlier in the manuscript.

Response: revised to “In accordance with other literature, MAB and MAC induce mitochondrial damage in macrophages that result in induced deficits in integrity of both mtDNA and MMP.”

Comment: Line 618: Please either consider removing “and immune cells” or adding “perhaps other” before it.

Response: Deficits in energy production and mitochondrial activity negatively impact the ability of macrophages and perhaps other immune cells to effectively combat bacterial infections.

Comment: Lines 631-632: Please consider minor revision for clarity.

Response: this has been revised, “ In addition to control over mitochondrial biogenesis, PGC-1α exerts multiple beneficial effects in immune cells, including expression of cytoprotective and antioxidant proteins.

Comment: Lines 650-651: Please consider revision/breaking the sentence.

Response: this has been revised, Our data show that NTM infections in vitro downregulate both PGC-1α and TFAM, two critical determinants of mitochondrial function that can impact the immune function of macrophages.

Comment: Lines 672-674: Please consider revision for better clarity.

Response:Sentences were revised to: “Metformin has been shown to reduce the severity of bacterial infections in preclinical and clinical retrospective studies, although the exact mechanisms remain elusive.The potential mechanisms by which Metformin exerts beneficial effects and rescues mitochondrial bioenergetics are related to its activation of AMPK and downstream activation of PGC-1α.”

Comment: Line 691: Please consider adding “and” after “bioenergetics,”.

Response: The wording was modified to read “In conclusion, our study shows that NTM infected macrophages exhibit mitochondrial damage with impaired bioenergetics and increased ROS production.”

---

## [Decision Letter · Decision Letter 2]

19 Jun 2024

PONE-D-23-34313R2PGC-1α Activation to Enhance Macrophage Immune Function in Mycobacterial InfectionsPLOS ONE

Dear Dr. Sadikot,

Thank you for submitting your manuscript to PLOS ONE. After careful consideration, we feel that it has merit but does not fully meet PLOS ONE’s publication criteria as it currently stands. Therefore, we invite you to submit a revised version of the manuscript that addresses the points raised during the review process.

Please submit your revised manuscript by Aug 03 2024 11:59PM. If you will need significantly more time to complete your revisions, please reply to this message or contact the journal office at plosone@plos.org. Please include the following items when submitting your revised manuscript:A rebuttal letter that responds to each point raised by the academic editor and reviewer(s). You should upload this letter as a separate file labeled 'Response to Reviewers'.A marked-up copy of your manuscript that highlights changes made to the original version. You should upload this as a separate file labeled 'Revised Manuscript with Track Changes'.An unmarked version of your revised paper without tracked changes. You should upload this as a separate file labeled 'Manuscript'.If applicable, we recommend that you deposit your laboratory protocols in protocols.io to enhance the reproducibility of your results. Protocols.io assigns your protocol its own identifier (DOI) so that it can be cited independently in the future. For instructions see: https://journals.plos.org/plosone/s/submission-guidelines#loc-laboratory-protocols. Additionally, PLOS ONE offers an option for publishing peer-reviewed Lab Protocol articles, which describe protocols hosted on protocols.io. Read more information on sharing protocols at https://plos.org/protocols?utm_medium=editorial-email&utm_source=authorletters&utm_campaign=protocols.

We look forward to receiving your revised manuscript.

Kind regards,

Frederick Quinn

Academic Editor

PLOS ONE

Journal Requirements:

Reviewers' comments:

Reviewer's Responses to Questions

**Comments to the Author**

1. If the authors have adequately addressed your comments raised in a previous round of review and you feel that this manuscript is now acceptable for publication, you may indicate that here to bypass the “Comments to the Author” section, enter your conflict of interest statement in the “Confidential to Editor” section, and submit your "Accept" recommendation.

Reviewer #2: (No Response)

2. Is the manuscript technically sound, and do the data support the conclusions?

Reviewer #2: Yes

3. Has the statistical analysis been performed appropriately and rigorously? 

Reviewer #2: N/A

4. Have the authors made all data underlying the findings in their manuscript fully available?

Reviewer #2: Yes

5. Is the manuscript presented in an intelligible fashion and written in standard English?

Reviewer #2: Yes

6. Review Comments to the Author

Reviewer #2: Few issues:

1. Ethics section, lines 320-322 of the first revised version has been omitted from the second (current) revised version. Please clarify.

2. Line 355 of the current version (ppargc1a) should be in italic

3. Line 357 of the current version (tfam) should be in italic

4. Figure 2 (2A to 2L) corresponding to data related to ZLN005 is missing.

7. PLOS authors have the option to publish the peer review history of their article (what does this mean?). If published, this will include your full peer review and any attached files.

Reviewer #2: No

---

## [Author Response · Author response to Decision Letter 2]

21 Jul 2024

We appreciate the reviewers detailed review and comments for correction. We have incorporated all the changes. 

Reviewer Two

Comment

1. Ethics section, lines 320-322 of the first revised version has been omitted from the second (current) revised version. Please clarify.

Response

The studies do not have any human specimens therefore the ethics section is omitted

Comment

2. Line 355 of the current version (ppargc1a) should be in italic

Response

This has been reformatted

Comment

3. Line 357 of the current version (tfam) should be in italic

Response: 

This has been formatted

Comment:

4. Figure 2 (2A to 2L) corresponding to data related to ZLN005 is missing.

Response:

ZLN005 has been added

---

## [Decision Letter · Decision Letter 3]

20 Aug 2024

PONE-D-23-34313R3PGC-1α Activation to Enhance Macrophage Immune Function in Mycobacterial InfectionsPLOS ONE

Dear Dr. Sadikot,

Thank you for submitting your manuscript to PLOS ONE. After careful consideration, we feel that it has merit but does not fully meet PLOS ONE’s publication criteria as it currently stands. Therefore, we invite you to submit a revised version of the manuscript that addresses the points raised during the review process.

Please submit your revised manuscript by Oct 04 2024 11:59PM. If you will need significantly more time to complete your revisions, please reply to this message or contact the journal office at plosone@plos.org. Please include the following items when submitting your revised manuscript:A rebuttal letter that responds to each point raised by the academic editor and reviewer(s). You should upload this letter as a separate file labeled 'Response to Reviewers'.A marked-up copy of your manuscript that highlights changes made to the original version. You should upload this as a separate file labeled 'Revised Manuscript with Track Changes'.An unmarked version of your revised paper without tracked changes. You should upload this as a separate file labeled 'Manuscript'.If applicable, we recommend that you deposit your laboratory protocols in protocols.io to enhance the reproducibility of your results. Protocols.io assigns your protocol its own identifier (DOI) so that it can be cited independently in the future. For instructions see: https://journals.plos.org/plosone/s/submission-guidelines#loc-laboratory-protocols. Additionally, PLOS ONE offers an option for publishing peer-reviewed Lab Protocol articles, which describe protocols hosted on protocols.io. Read more information on sharing protocols at https://plos.org/protocols?utm_medium=editorial-email&utm_source=authorletters&utm_campaign=protocols.

We look forward to receiving your revised manuscript.

Kind regards,

Frederick Quinn

Academic Editor

PLOS ONE

Journal Requirements:

Reviewers' comments:

Reviewer's Responses to Questions

**Comments to the Author**

1. If the authors have adequately addressed your comments raised in a previous round of review and you feel that this manuscript is now acceptable for publication, you may indicate that here to bypass the “Comments to the Author” section, enter your conflict of interest statement in the “Confidential to Editor” section, and submit your "Accept" recommendation.

Reviewer #2: (No Response)

2. Is the manuscript technically sound, and do the data support the conclusions?

Reviewer #2: Yes

3. Has the statistical analysis been performed appropriately and rigorously? 

Reviewer #2: N/A

4. Have the authors made all data underlying the findings in their manuscript fully available?

Reviewer #2: Yes

5. Is the manuscript presented in an intelligible fashion and written in standard English?

Reviewer #2: Yes

6. Review Comments to the Author

Reviewer #2: Although this manuscript is interesting and worthy of publication, it appears that the manuscript is not carefully prepared evident by multiple errors over several revisions which has unfortunately continued even in the most recent version.

Figure 2, is not incorporated into the figures and is separately added out of place at the end of the figures. The quality of the figure 2 is not great and apparently does not have a corresponding TIFF file (or at least this reviewer could not download it). Therefore, as is, figure 2 is not suitable for publication. Please make sure that a good quality figure is added and properly incorporated in its right place among figures

Line 363: It seems that “and TFAM (FIG 2F)” should be “TFAM (FIG 2G)”

Line 367: It seems that “PGC-1a (FIG 2G)” should be “PGC-1a (FIG 2F)”

Please make the above correction and make sure that next version is thoroughly reviewed and does not have any more errors.

Thank you

7. PLOS authors have the option to publish the peer review history of their article (what does this mean?). If published, this will include your full peer review and any attached files.

Reviewer #2: No

---

## [Author Response · Author response to Decision Letter 3]

22 Aug 2024

All requested changes have been incorporated into the manuscript and figure files.

Reviewer Two

Comment:

Reviewer #2: Although this manuscript is interesting and worthy of publication, it appears that the manuscript is not carefully prepared evident by multiple errors over several revisions which has unfortunately continued even in the most recent version.

Figure 2, is not incorporated into the figures and is separately added out of place at the end of the figures. The quality of the figure 2 is not great and apparently does not have a corresponding TIFF file (or at least this reviewer could not download it). Therefore, as is, figure 2 is not suitable for publication. Please make sure that a good quality figure is added and properly incorporated in its right place among figures

Response:

Figure 2 was reformatted to a higher quality TIFF file and uploaded in the proper sequence.

Comment:

Line 363: It seems that “and TFAM (FIG 2F)” should be “TFAM (FIG 2G)”

Response:

The proper figure title was added

Comment:

Line 367: It seems that “PGC-1a (FIG 2G)” should be “PGC-1a (FIG 2F)”

Response:

The proper figure title was added

---

## [Decision Letter · Decision Letter 4]

9 Sep 2024

PGC-1α Activation to Enhance Macrophage Immune Function in Mycobacterial Infections

PONE-D-23-34313R4

Dear Dr. A little ,

We’re pleased to inform you that your manuscript has been judged scientifically suitable for publication and will be formally accepted for publication once it meets all outstanding technical requirements.

Kind regards,

Frederick Quinn

Academic Editor

PLOS ONE

Additional Editor Comments (optional):

Reviewers' comments:

Reviewer's Responses to Questions

**Comments to the Author**

1. If the authors have adequately addressed your comments raised in a previous round of review and you feel that this manuscript is now acceptable for publication, you may indicate that here to bypass the “Comments to the Author” section, enter your conflict of interest statement in the “Confidential to Editor” section, and submit your "Accept" recommendation.

Reviewer #2: (No Response)

2. Is the manuscript technically sound, and do the data support the conclusions?

Reviewer #2: Yes

3. Has the statistical analysis been performed appropriately and rigorously?

Reviewer #2: N/A

4. Have the authors made all data underlying the findings in their manuscript fully available?

Reviewer #2: Yes

5. Is the manuscript presented in an intelligible fashion and written in standard English?

Reviewer #2: Yes

6. Review Comments to the Author

Reviewer #2: Acceptable for publication with following corrections:

- Line 406: "FIG 3F" to "FIG 3G"

- Line 409: "FIG 3G" to "FIG 3F"

7. PLOS authors have the option to publish the peer review history of their article (what does this mean?). If published, this will include your full peer review and any attached files.

Reviewer #2: No

---

## [Editor Report · Acceptance letter]

28 Nov 2024

PONE-D-23-34313R4 

PLOS ONE

Dear Dr. Sadikot, 

I'm pleased to inform you that your manuscript has been deemed suitable for publication in PLOS ONE. Congratulations! Your manuscript is now being handed over to our production team.

Kind regards, 

on behalf of

Dr. Frederick Quinn 

Academic Editor

PLOS ONE